# Performing an Ultrasound-Guided Percutaneous Needle Kidney Biopsy: An Up-To-Date Procedural Review

**DOI:** 10.3390/diagnostics11122186

**Published:** 2021-11-24

**Authors:** Antonio Granata, Giulio Distefano, Francesco Pesce, Yuri Battaglia, Paola Suavo Bulzis, Massimo Venturini, Stefano Palmucci, Vito Cantisani, Antonio Basile, Loreto Gesualdo

**Affiliations:** 1Nephrology and Dialysis Unit, “Cannizzaro” Hospital, 95126 Catania, Italy; antonio.granata4@tin.it; 2Radiology Unit 1, Department of Medical Surgical Sciences and Advanced Technologies “GF Ingrassia”, University Hospital “Policlinico-San Marco”, University of Catania, 95123 Catania, Italy; spalmucci@sirm.org (S.P.); basile.antonello73@gmail.com (A.B.); 3Nephrology, Dialysis and Transplantation Unit, Department of Emergency and Organ Transplantation, University of Bari “Aldo Moro”, 70121 Bari, Italy; f.pesce81@gmail.com (F.P.); p.suavobulzis@gmail.com (P.S.B.); l.gesualdo@nephro.uniba.it (L.G.); 4Division of Nephrology and Dialysis, St. Anna University Hospital, 44121 Ferrara, Italy; battagliayuri@gmail.com; 5Department of Diagnostic and Interventional Radiology, Ospedale di Circolo e Fondazione Macchi, University of Insubria, 21100 Varese, Italy; massimo.venturini@uninsubria.it; 6Department of Radiology, Policlinico Umberto I, Sapienza Rome University, 00161 Rome, Italy; vito.cantisani@uniroma1.it

**Keywords:** kidney, ultrasound, biopsy, chronic kidney disease, acute kidney disease, proteinuria, hematuria, percutaneous, ultrasound-guided

## Abstract

Ultrasound-guided percutaneous renal biopsy (PRB) has revolutionized the clinical practice of nephrology in the last decades. PRB remains an essential tool for the diagnosis, prognosis, and therapeutic management of several renal diseases and for the assessment of renal involvement in systemic diseases. In this study, we examine the different applications and provide a review of the current evidence on the periprocedural management of patients. PRB is recommended in patients with significant proteinuria, hematuria, acute kidney injury, unexpected worsening of renal function, and allograft dysfunction after excluding pre- and post-renal causes. A preliminary ultrasound examination is needed to assess the presence of anatomic anomalies of the kidney and to identify vessels that might be damaged by the needle during the procedure. Kidney biopsy is usually performed in the prone position on the lower pole of the left kidney, whereas in patients with obesity, the supine antero-lateral position is preferred. After preparing a sterile field and the injection of local anesthetics, an automatic spring-loaded biopsy gun is used under ultrasound guidance to obtain samples of renal parenchyma for histopathology. After the procedure, an ultrasound scan must be performed for the prompt identification of potential early bleeding complications. As 33% of complications occur after 8 h and 91% occur within 24 h, the ideal post-procedural observation time is 24 h. PRB is a safe procedure and should be considered a routine part of the clinical practice of nephrology.

## 1. Introduction

Nephrology was revolutionized in the 1950s and 1960s by the introduction of percutaneous renal biopsy (PRB) into clinical practice. The first attempt to perform a percutaneous needle biopsy of the kidney was described in 1951 by Iversen and Brun, who used a fluoroscopically guided approach and obtained parenchyma samples after antegrade pyelography in a sitting position, with a success rate of only 40% [1]. The technique was modified by Karm, who, in 1954, proposed the performance of the biopsy in the prone position with an ad hoc needle, reaching success in 96% of cases [2]. Since the 1980s, percutaneous biopsy has been performed under ultrasound guidance using spring-loaded biopsy needles, mainly in nephrological and radiological departments. Despite its unique role in the management of nephrological diseases, it has been reported that about half of all post-graduate nephrology program directors do not consider biopsies to be an essential component of the core curriculum for residents [3]. In this review, we provide an operative guide for the safe and successful execution of ultrasound-guided renal biopsies for clinical nephrologists and radiologists.

## 2. Indications

All physicians must know the indications for a renal biopsy and should be aware of the contraindications and risks in order to estimate the risk/benefit ratio of the procedure [4]. Common indications include significant proteinuria (defined as >1 g/day in the absence of other comorbidities or rapid increases in proteinuria in patients with a diagnosis of kidney disease, systemic immunological disease, or diabetes), acute kidney injury (AKI) when an intrinsic etiology is suspected, hematuria associated with proteinuria and/or decreased renal function or suspected secondary glomerular disease, rapid, and unexpected worsening of eGFR, or rapid increase in proteinuria in known chronic kidney disease (CKD). A renal biopsy is also essential to assess the extent of renal damage in systemic diseases, such as systemic lupus erythematosus or paraproteinemia, due to the impact on treatment (Table 1). In patients without already known glomerular or systemic pathologies, isolated hematuria, or proteinuria <1 g/day does not necessarily require biopsy, as the course of these nephropathies is generally indolent and the risk/benefit ratio of knowing the histological diagnosis does not justify an invasive procedure [5,6].

## 3. Contraindications

Coagulation impairments are, in some cases, the main contraindication for PRB. Currently, there is no consensus on the management of patients on aspirin. Some centers regularly discontinue aspirin within seven days before and two days after renal biopsy. The discontinuation of aspirin, however, does not appear to reduce the risk of major bleeding complications [6]. Other antiplatelet agents (e.g., clopidogrel) are almost always discontinued at least one week before the biopsy. Therefore, it is important to promptly contact nephrologists for any patient who is a candidate for renal biopsy thus that the procedure is not delayed if these agents are not stopped in a timely manner. Non-correctable coagulopathy and lack of safe access are two absolute contraindications of biopsy performance, which is considered a relatively invasive procedure; an uncooperative or unwell patient is another contraindication to consider [7,8]. The alterations in coagulation parameters may require postponing the procedure. Since renal biopsy is considered a high-risk bleeding procedure, INR and a platelet count are routinely recommended, while aPTT is required if the patient undergoes unfractionated IV heparin therapy. An INR >1.5 or platelet counts below 50,000/× 10^9^/L are also contraindications to performing the procedure but are virtually susceptible to correction [9]. The same contraindications reported for the native kidney are valid for the transplanted kidney. Additional contraindications include urosepsis, peri-renal abscesses, hydronephrosis, renal malformations (horseshoe kidney and native solitary kidney), multiple bilateral cysts, and uncontrolled high blood pressure [6] (Table 2).

## 4. Management of Relative Contraindications

Although the relative risk of severe bleeding is not significantly lowered by the withdrawal of antiplatelet drugs (aspirin and clopidogrel), it is currently accepted to discontinue the drug within five days prior to the procedure. New oral anticoagulant drugs must be suspended at least three days before, and they can be temporarily replaced by low-molecular-weight heparin (LMWH). Dicoumarols should be discontinued approximately five days prior to biopsy and replaced by low-molecular-weight heparin (LMWH) until the next day (acenocumarolo) or two days after (warfarin). LMWH administration should be stopped 12 h before biopsy if it is administered for prophylactic purposes, while it should be discontinued 24 h earlier if administered at full doses for therapeutic purposes [8,9,10,11]. LMWH administration can be resumed after 12 h (prophylactic dose) or 24–72 h (full dose) [11]. The targets for systolic and diastolic blood pressure are <140 mmHg and <90 mmHg, respectively, as it has been shown that the risk of complications (mainly bleeding) is 10 times higher if blood pressure exceeds these thresholds and, up to 26 times higher, if the systolic exceeds 170 mmHg [6]. All oral antihypertensive medications that the patient is currently taking should be continued regularly and may possibly be supplemented by IV therapy immediately prior to the procedure.

## 5. Equipment

A table covered with a sterile cloth should be set with the following tools: drapes, disinfectant sponges, sterile gauze pads, surgical scalpel blade no. 11, anesthetic syringe, spinal anesthesia needle, and an automatic or semi-automatic spring-loaded biopsy system. A 15/19 cm long needle with a penetration depth of 22 mm and a sample notch of 13–18 mm is adequate for the purpose; 14-, 16-, and 18-gauge needles can be used, and a 16-gauge needle appears to be associated with the best risk/benefit ratio in terms of glomeruli harvested and severe bleeding. Some companies market biopsy kits with all of the essential tools for performing the procedure (Figure 1) [12,13,14]. The operator and their assistant must be provided with sterile gloves, eye protection, surgical caps, and surgical masks. A full-size sterile drape should be placed over the patient after the access site skin has been sterilized with 7.5% povidone-iodine solution. An ultrasound machine with convex-array, high-resolution, Doppler module is preferred for this procedure (Figure 2). Sterile transduction gel, an acoustically transparent sterile transducer sheath, software with guide marks with an inclination of 20–30°, and sterile rubber bands or clips to secure the sheath around the transducer are also needed (Table 3).

The specimen should be immediately evaluated using an optical microscope before preparing it for the pathologist to decide how many biopsy passes are needed to obtain an adequate renal sample. In general, a sampling of 20 glomeruli is considered adequate to allow for a correct diagnosis [4].

## 6. Preparation

Before carrying out any invasive procedure, it is necessary to obtain informed consent from the patient or whoever has legal responsibility, in accordance with national regulations and institutional guidelines. Ideally, written consent should be obtained the day before the procedure and possibly confirmed on the day of the procedure; however, this timing is often waived. The patient must be informed about the indication for carrying out the procedure, the potential complications, and the benefits with respect to choosing the optimal treatment after a correct diagnosis [15]. Patients should have a light meal on the evening before the procedure, and the administration of a cleansing enema can be considered. On the day of the procedure, patients must be fasting. Any antihypertensive drugs should be given regularly. A peripheral vein must be cannulated. Blood pressure should be measured before performing PRB. The bladder needs to be empty. The usefulness of antibiotic prophylaxis in procedures where one does not cross-contaminated territories is debated, and it is not clear which antibiotic to choose or for how long to prescribe it. However, it is considered appropriate to administer a dose of broad-spectrum antibiotics before starting the procedure.

## 7. Position of the Patient

A kidney biopsy is usually performed in the prone position on the lower pole of the left kidney, which is preferred to reduce the risk of inadvertent injury to a major vessel. If the patient is obese or has breathing difficulties, a supine anterolateral position (SALP) is more suitable and does not increase the risk of complications [10].

## 8. Prebiopsy Ultrasound Study

Before performing an ultrasound-guided PRB, it is advisable to perform a complete ultrasound examination of the kidneys in the same position as the procedure to evaluate the location of the kidney, its excursion during breathing, the cortical thickness, and the distance of the kidney from the skin surface. These data are useful to plan the procedure. Furthermore, this allows for the exclusion of anatomical variants (for example, horseshoe kidney), hydronephrosis, and cystic lesions, which may be contraindications for the procedure [16]. The examination must be completed with an ECD assessment to rule out the presence of vessels along the presumed path of the needle (Figure 3). Vascular abnormalities are reported in up to 10% of patients who undergo renal biopsy, and this may explain the reduction in complications observed when an ECD study preceded the procedure [15].

## 9. Procedure

### 9.1. Approach

An ultrasound-guided PRB can be performed either freehand or with the aid of needle supports to force its trajectory (needle-guided insertion). Compared with the probe, the needle can have a parallel or transverse insertion: the guided needle technique can only be used with parallel insertion [3]. In most cases, the prone position is preferred; in patients who are obese or pregnant (after the 20th week), a lateral decubitus or sitting upright position can be evaluated [9]. The use of needle guides was associated with greater reproducibility and speed of execution compared with the freehand technique, especially for beginners, but there were no significant differences in terms of complications [17].

It is preferable to perform the ultrasound-guided needle biopsy on the lower pole of the left kidney, at the level of Brodel’s line (a relatively poorly vascularized plane between the anterior and posterior branches of the renal artery). The inferior pole of the left kidney is in fact, more accessible than the contralateral, and it is more distant from large-caliber vessels [18].

### 9.2. Performing the Biopsy in the Prone Position

The patient is placed in a prone position, possibly with a pillow under the abdomen to reduce the physiological dorsal lordosis and to make the kidney more accessible (Figure 4). Patients must be collaborative, and this is an important aspect in the execution of an ultrasound-guided needle biopsy, given the motion of the kidneys during respiration. The use of anxiolytic drugs before the procedure can be considered. However, in uncooperative or seriously ill patients, the use of hypnotic drugs might require the support of an anesthesiologist [7,19]. The whole procedure must be carried out under sterile conditions. First, it is necessary to cleanse the skin with povidone-iodine and then to apply the sterile sheets to obtain a sterile window on the access site; the probe must be covered by sterile protection, and the ultrasound examination must be carried out using a sterile gel. An additional preprocedural ultrasound scan allows for the confirmation of the planned position and trajectory. The trajectory of the needle through the guide can be displayed on the ultrasound monitor.

Local anesthesia is performed (Appendix A). Under ultrasound guidance, the biopsy needle is then introduced with an inclination of about 20° or 30° through the subcutaneous tissue, the muscular planes, and the fascia until it stops right above the renal capsule (Figure 5, Appendix A). Although puncture of the cortical region of the kidney is always performed in B-mode, an ECD scan is advisable to identify abnormal vessels (Appendix A). In that case, it is appropriate to perform the biopsy on the contralateral kidney.

It is important to underline that the needle and the tip must always be visualized on the monitor during the procedure to avoid errors (Appendix A). Using a small amount of saline solution or by shaking the needle, it is possible to increase the visibility of the tip [19]. The patient’s collaboration and breathing, particularly inspiratory apnea during the execution of the biopsy, are fundamental since unexpected movements of the kidney can shift the trajectory of the needle with potential unexpected and unwanted outcomes. When the optimal target is reached, with the needle resting on the renal capsule, the automatic needle is triggered, thus activating the descent, initially, of the mandrel and, subsequently, of the shirt, which locks up the sample.

The needle is then extracted from the kidney, and by opening it, the biopsy sample is removed and then either laid down on a gauze soaked in physiological solution or in a sterile container containing physiological solution. If available, an optical microscope can be used to evaluate fresh biopsy samples for the presence of cortical parenchyma. Usually, two passes are sufficient to obtain a renal parenchyma specimen sufficient for analysis by light microscopy, immunofluorescence, and electron microscopy, if necessary. In many institutions, the samples should be fixed in formalin solution before sending them to the pathologist. In any case, it is always advisable to send an accompanying clinical report to direct the diagnosis.

Once the needle has been extracted, hemostasis by compression is applied for a few minutes.

At the end of the procedure, before moving the patient, it is necessary to perform an additional US-ECD evaluation to check the absence of hematomas, pseudoaneurysms, or arteriovenous fistulas, which are the most common periprocedural adverse events. The patient should be informed that a period of bed rest is necessary for at least 24 h after the procedure.

### 9.3. Performing the Biopsy in the Supine Antero-Lateral Position (SALP)

This technique can be used in patients who are obese (BMI > 30) to avoid several inconveniences, such as poor respiratory performance due to the prone position, the depth of the inferior renal pole, and the difficulty of some patients in maintaining the position for a long time (Figure 6). This technique has a significantly lower complication rate than the classic approach, excellent diagnostic power, and better patient compliance in patients who are obese [20]. In the SALP position, a towel is placed under the ipsilateral shoulder and the gluteus to elevate the flank by an angle of 30°. The ipsilateral arm is placed over the thorax while the contralateral is abducted and used for intravenous perfusion. The ipsilateral leg is slightly flexed over a pillow, while the contralateral is flexed and abducted, thus that its lateral aspect is lying on the table. This position provides full exposure of Petit’s triangle, thus providing enough space to perform ultrasound scanning and to easily orient the ultrasound-guided puncture toward the inferior renal pole. After shaving and draping the flank, the kidney is ultrasound-scanned to determine the ideal puncture path. The identification of the lower kidney pole by ultrasound scanning is easy, and the quality of image resolution is similar to the prone position. After injecting a local anesthetic, an automatic needle is ultrasound-guided to the capsule in the lower pole of the kidney and fired into the renal parenchyma, similar to what is expected for biopsies performed in the supine position.

### 9.4. Percutaneous Renal Transplant Biopsy

A biopsy of transplanted kidney is usually easier to perform, since the graft is more superficial, more accessible, and relatively static with respiration. The graft is in the anterior part of the lower abdomen in the iliac region, and it lays over the iliopsoas muscle in the retroperitoneal space. The transplanted kidney is separated from the anterior abdominal wall only by the transversalis fascia, abdominal muscles, and subcutaneous tissue: the supine position with an approach from the anterolateral surface of the abdomen generally allows for easy access to the organ. The steps necessary to obtain an ultrasound-guided needle biopsy of graft are quite similar to those described for the native kidney, but it must be noted that the risk of complications is higher. If the biopsy is performed a few days after the transplant, there is a high rate of complications such as relevant hemorrhages and lesions of the parenchyma due to the frailty of the kidney after cold ischemia. On the other hand, a long-transplanted kidney instead has pericapsular fibrosis due to the chronic desmogenic reaction of the perirenal tissues. The fibrosis surrounding the transplanted organ could make it difficult to penetrate the renal capsule. In these cases, it is advisable to go deeper, with the tip of the needle a few millimeters below the capsular plane, and only after that is it advisable to activate the automatic needle system. In the case of suspected rejection, samples from different spots of the kidney should be obtained; as the inflammatory lesions may have a heterogeneous distribution, the trajectory within the renal cortex should be slightly oblique to increase the quantity of cortical structures removed for analysis and, at the same time, to reduce the risk of vessel injury [16].

## 10. Monitoring

A consolidated literature review shows that 3% of complications occur within 8 h and that 91% occur within 24 h [19]. The monitoring period must, therefore, extend to the day following the procedure, and during this period, the patient must remain enticed. In the first 6 h after the procedure, it is necessary to measure the blood pressure every hour. A check of the blood count should be performed after 4–6 h and after 24 h. It is important to point out that sudden and intense low back pain, abdominal pain, or a sudden blood pressure drop can be a symptom of massive bleeding, thus it is necessary to repeat a blood count immediately, to perform a US examination and to eventually perform a CT scan. A check with Doppler ultrasound must be performed on the following day. The patient should be advised to rest at home for 7 days, avoiding car trips and abstaining from intense physical activity for two weeks.

## 11. Special Population

A few considerations should be pointed out in some categories of patients, such as patients who are older, pregnant women, and those with renal dysmorphism or a solitary kidney. Although CKD is not infrequent in patients who are older, the registers show that only a small percentage undergo a diagnostic assessment with biopsy [21]. Evidence in the literature demonstrates that, although there are more minor complications in patients >61 years old than in the rest of the population, the rate of severe complications (including bleeding treated surgically or with intravascular embolization) does not significantly increase [22]. Delaying or not suggesting PRB based on the patient’s age is, therefore, not justified.

In addition to the classic indications for performing a kidney biopsy, a pregnant woman may present with AKI in the setting of eclampsia. The prone position, more commonly used for the general population, is not feasible in pregnant women, for whom a SALP is recommended. It has been reported that, during pregnancy, the incidence of major adverse events (including in this case also pre-term delivery) is 2% of the total PRB, with the highest incidence between 23 and 26 weeks of gestation [23].

A solitary kidney is considered a relative contraindication to PRB by some authors. In reality, there is no evidence in the literature of a significantly higher number of complications in this population, and fear in this sense is not justifiable. In patients with renal dysmorphism, in particular, in the so-called horseshoe kidney situation, particular attention must be paid to possible associated vascular anomalies. Since abnormal vessels could be in the path of the needle, it is mandatory to perform an ECD examination during the procedure [16].

## 12. Complications

Although the ultrasound-guided execution of PRB after a thorough clinical assessment is relatively safe, adverse events are not infrequent and can be potentially fatal. Bleeding is certainly the most common complication. Hematoma (75% of complications) is generally well-recognizable with post-procedural ultrasound. It can be subcapsular, retroperitoneal or, rarely due to injury of the lumbar vessels (Figure 7). It has been described that the majority of patients with a detectable hematoma on the control ultrasound had minor or major complications (including bleeding requiring transfusion or radiologist intervention) in the few hours following the procedure. Hematoma is, therefore, a predictor of bleeding-related complications, but it seems as if there is no significant relationship between its size and the extent of bleeding [24,25]. Symptoms range from completely silent (small subcapsular hematomas) to hematuria, flank pain, anemia, and shock. The control of blood pressure and local compression may be sufficient in the management of subcapsular bleeding, while in the most severe cases, a super-selective angiography of the arterial vessels involved and subsequent embolization must be used [6]. Surgical treatment (which may include nephrectomy in fewer than 1% of complicated cases) is reserved for cases that cannot be treated intravascularly or cases of hemodynamic instability [7]. A special case of subcapsular bleeding is the page kidney, which is caused by the accumulation of blood in the perinephric or subcapsular space, resulting in extrinsic compression of the involved kidney, renal ischemia, activation of the renin–angiotensin–aldosterone system, and systemic hypertension. Arteriovenous fistula (AVF) is a relatively rare complication of kidney biopsy detectable with ECD, which is especially common in transplanted kidneys (described in up to 10% of biopsies). The treatment can be conservative or interventional in confirmed cases.

## 13. Conclusions

Ultrasound-guided renal needle biopsy is an essential tool in daily nephrological practice, necessary to provide a histological diagnosis with a possible impact on therapeutic management. The technique of this invasive procedure is highly standardized, and it has been shown to be associated with a high technical success rate and a relatively small number of minor complications, in addition to major complications being quite rare. The technique for performing this minimally invasive procedure is highly standardized, and in expert hands, it has a high success rate and relatively few manageable complications. The safety of the procedure relies on the correct selection of patients, a good knowledge of the tools, proper management of modifiable risk factors, and subsequent patient monitoring. In our opinion, given its essential role in the management of renal parenchymal diseases, knowledge of the indications and procedural method of PRB must be clear to all clinical nephrologists and radiologists who deal with interventional procedures, reserving the intervention of second-level centers for the treatment of specific populations with potentially greater risk of complication.

## Figures and Tables

**Figure 1 diagnostics-11-02186-f001:**
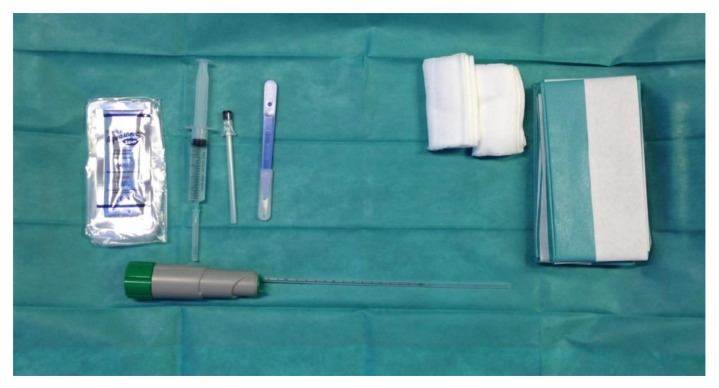
Commercially available renal biopsy kit, consisting of an automatic spring-loaded biopsy system with an 18 G needle and a 1 or 2 cm slit, sterile drapes, sterile gauze, sterile ultrasound gel, a syringe and needle for anesthesia, a scalpel, and a swab for surface disinfection.

**Figure 2 diagnostics-11-02186-f002:**
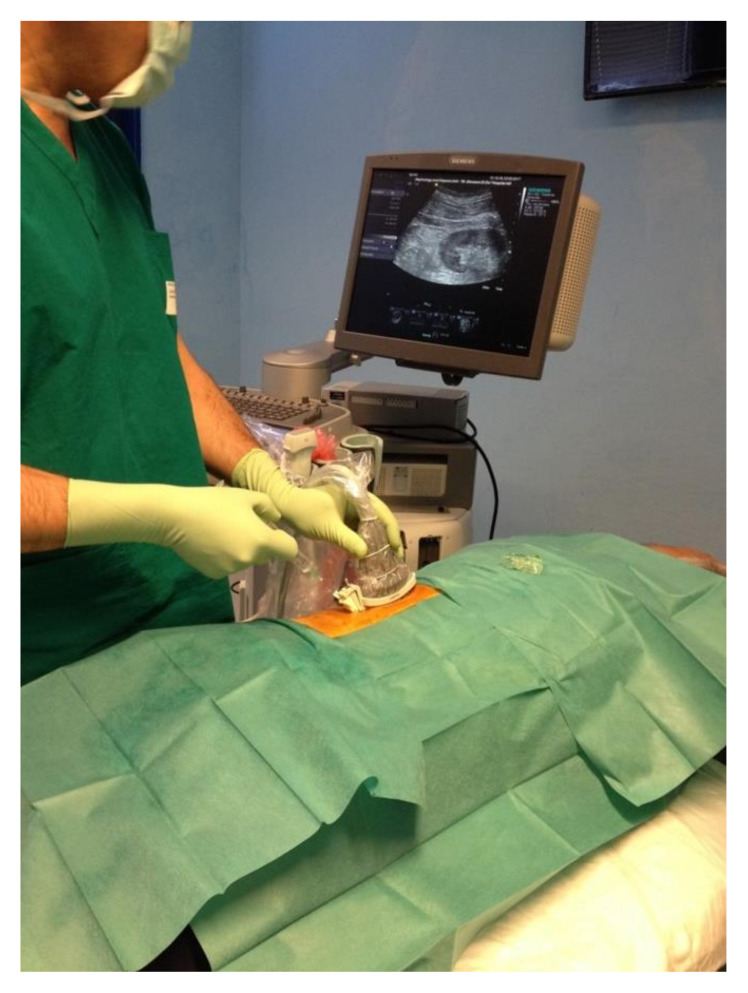
The correct position of the operator with respect to the patient is demonstrated. A needle guide system was mounted on the convex probe. The whole procedure is conducted in a sterile state, and the sterile sheets surrounding the uncovered skin areas are not punctured.

**Figure 3 diagnostics-11-02186-f003:**
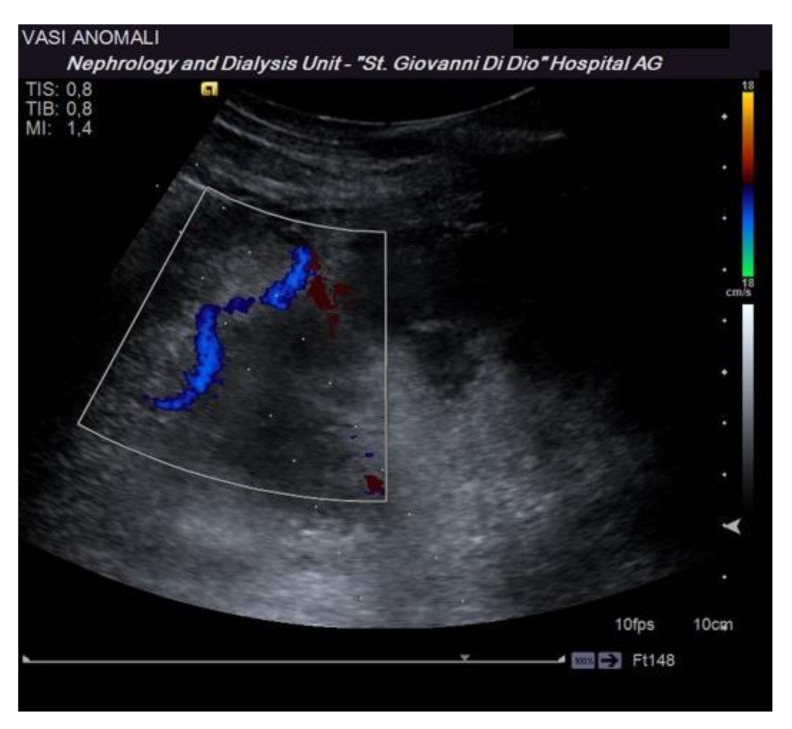
The dotted lines show the expected trajectory of the needle when correctly fixed to the appropriate support. At the lower pole of the kidney, the site of the biopsy, there is an abnormal vascular formation that would have been crossed by the needle if it had not been recognized.

**Figure 4 diagnostics-11-02186-f004:**
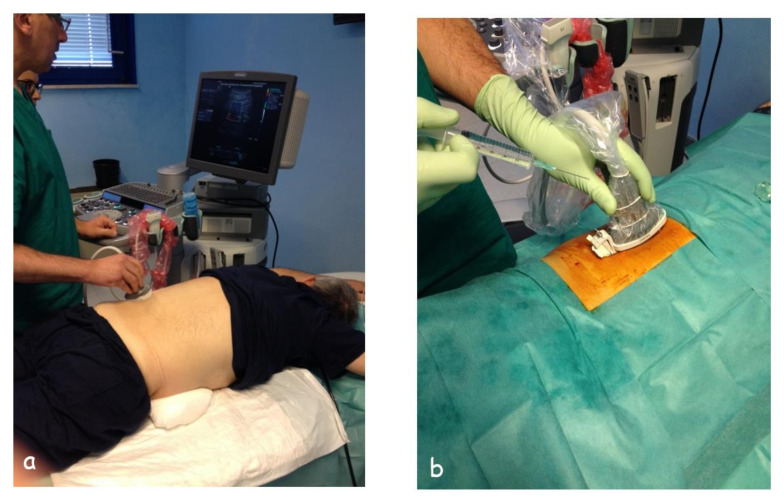
(**a**) Demonstration of the correct position of a patient undergoing a renal biopsy in prone position, the position in which the biopsy is performed must also be maintained during the ultrasound examination, and (**b**) the same patient after the preparation of a sterile field and application of the drapes, just before the injection of the anesthetic.

**Figure 5 diagnostics-11-02186-f005:**
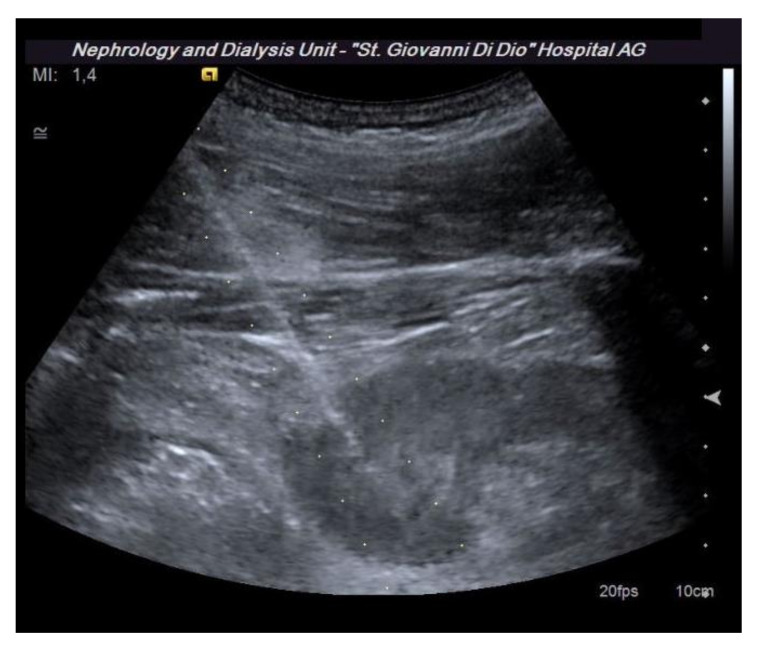
US highlights the trajectory of the needle that crosses the capsule with the tip in the internal region of the cortex. This image shows the needle after the activation of the trigger mechanism. The operator must verify that the needle does not reach the region of the renal pelvis, considering both the trajectory and the maximum possible excursion of the tip after the activation of the trigger mechanism.

**Figure 6 diagnostics-11-02186-f006:**
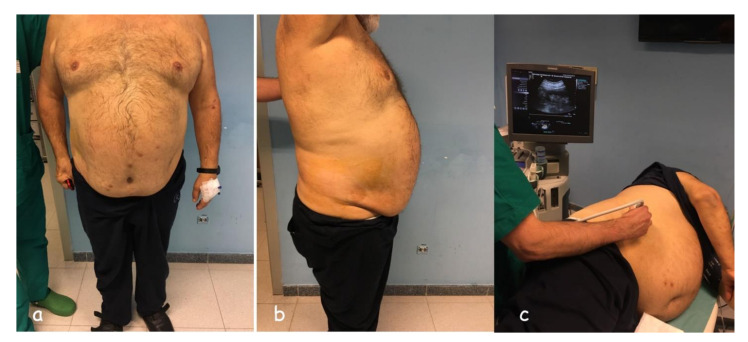
Supine antero-lateral position (SALP) is indicated in patients who are obese (**a**), (**b**); (**c**) pre-bioptic study.

**Figure 7 diagnostics-11-02186-f007:**
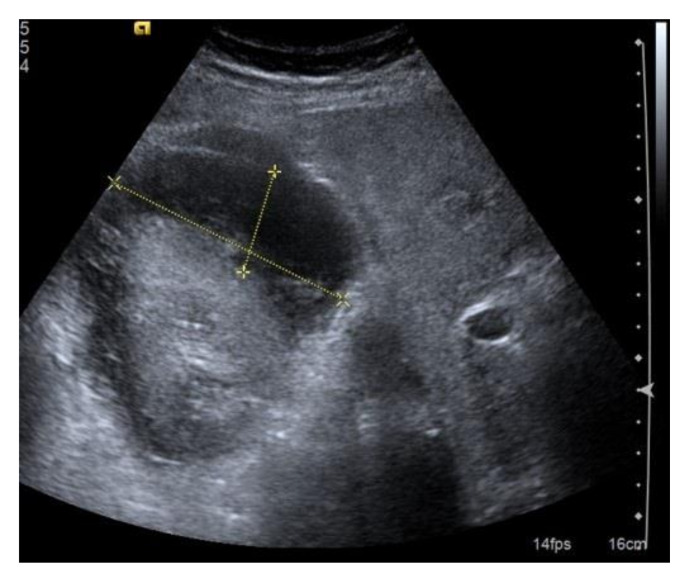
Extensive post-bioptic subcapsular hematoma; the anecogenicity of the effusion indicates that the bleeding is recent, but it is not possible to obtain reliable information as to whether bleeding still exists. Size is an unreliable parameter in these cases. In the presence of post-biopsy hematoma, non-invasive dynamic contrast studies (CEUS or contrast-enhanced CT) can be very useful to evaluate a bleeding source and any rarer post-biopsy vascular complications.

**Table 1 diagnostics-11-02186-t001:** Indications for performing renal biopsy [5,6].

Proteinuria >1 g/day (new finding, in the absence of other comorbidities), rapid increasing of proteinuria in a patient with known kidney disease, systemic immunological disorder or diabetes.
Rapid increase in proteinuria (in already known kidney disease, systemic immunological disease, or diabetes).
Acute kidney injury.
Rapid and unexpected worsening of eGFR in already known kidney disease.
Hematuria is associated with proteinuria and/or decreased renal function.
Systemic lupus erythematosus and paraproteinemia with any renal involvement.

**Table 2 diagnostics-11-02186-t002:** Absolute and relative contraindications for performing renal biopsy [6,7,8].

Absolute	Relative
Non-correctable coagulopathy (INR >1.5).	Therapy with antiplatelet agents, heparins or oral anticoagulants (to be suspended before the procedure; see main text).
Lack of safe access.	Hypertension (management with oral or IV drugs before the procedure).
Platelet counts below 50,000/× 10^9^/L.	Abscess.
Non-correctable hypertension.	Renal malformation.

**Table 3 diagnostics-11-02186-t003:** Tools of the trade.

Disinfectant sponges/buffer and 7.5% povidone-iodine solution	Sterile gauze pads	Sterile drapes
Surgical scalpel blade no. 11	Anesthetic syringe with spinal anesthesia needle	Personal protective equipment, sterile gloves, and apron
Sterile set for ultrasound probe (with probe cover, sterile gel, rubber bands)	Containers with formalin for fixing the histological samples taken	Automatic or semi-automatic spring-loaded biopsy system

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
