# Peer review of "Performing an Ultrasound-Guided Percutaneous Needle Kidney Biopsy: An Up-To-Date Procedural Review"

_diagnostics, 2021, doi:10.3390/diagnostics11122186_

Round 1

Reviewer 1 Report

This is a good review on ultrasound-guided percutaneous renal biopsy. It is well written. However, a careful edit is still needed, especially for texts in lines 43-44, 45. The conclusion part may be improved with specific suggestions or author’s experiences in ultrasound-guided percutaneous renal biopsy.

Author Response

We thank the Reviewer for the positive comments and encouragement to make some useful improvements. We followed all the suggestions as we have completely re-edited the manuscript improving the overall readability. We feel that these changes meet the indications of the Reviewer. Lines 43-44, 45 have been modified: we added a sentence that was not present in the draft. We improved the conclusions by commenting on our point of view based on our experience

Reviewer 2 Report

This paper is quite interesting for physicians with poor experience in renal biopsy, and can have some usefulness as a tutorial , but in my opinion it can not represent a true guideline. Therefore, the authors should modify the title.

English language strongly needs extensive revision, as there are several errors and in some cases wording is very macaronic. 

Fig. 5: in my opinion the path of the needle is not optimal, as it is directed to the medullary and renal pelvis; it would be better with opposite incline.

Fig. 7: the anechogenicity of post-biopsy hematoma suggests that the bleeding is quite recent; moreover, ultrasound can not provide reliable informations on the time of bleeding. Doppler-US and, more and more, CEUS can provide available informations.

Author Response

-This paper is quite interesting for physicians with poor experience in renal biopsy, and can have some usefulness as a tutorial , but in my opinion it can not represent a true guideline. Therefore, the authors should modify the title.

--R: We thank the Reviewer for the overall favorable comment. We were able to respond to all of the Reviewer's inquiries. As suggested, we changed the proposed title as follows: Performing an ultrasound guided percutaneous needle kidney biopsy: an up-to-date procedural review.

-English language strongly needs extensive revision, as there are several errors and in some cases wording is very macaronic.

--R: we have entirely re-edited the manuscript, we have corrected grammatical errors and we have rephrased some sentences using a more appropriate terminology, to improve the readability of the paper.

-Fig. 5: in my opinion the path of the needle is not optimal, as it is directed to the medullary and renal pelvis; it would be better with opposite incline.

--R: We carefully considered this Reviewer's comment in relation to the image we proposed. Actually, given the trajectory, the tip of the needle remains in the cortico-medullary region, safely, and does not reach the region of the renal pelvis. However, we agree with the Reviewer's point of view, and deemed appropriate to modify the caption as follows: "US highlights the trajectory of the needle that crosses the capsule with the tip in the internal region of the cortex; this image shows the needle after the activation of the trigger mechanism. The operator must verify that the needle does not reach the region of the renal pelvis, taking into account both the trajectory and the possible maximum excursion of the tip after the activation of the trigger mechanism".

-Fig. 7: the anechogenicity of post-biopsy hematoma suggests that the bleeding is quite recent; moreover, ultrasound can not provide reliable informations on the time of bleeding. Doppler-US and, more and more, CEUS can provide available informations.

--R: We have now modified the caption as follows: "Extensive post-bioptic subcapsular hematoma; the anecogenicity of the effusion indicates that the bleeding is recent, but it is not possible to obtain reliable information whether or not a bleeding still exists; size is an unreliable parameter in these cases. In the presence of post-biopsy hematoma, non-invasive dynamic contrast studies - CEUS or contrast-enhanced CT - can be very useful to evaluate a bleeding source and any further rarer post-biopsy vascular complications."

Round 2

Reviewer 2 Report

The authors satisfactorily answered my observations, and slightly improved english language. However, the language is still unacceptable is the present form, and strongly needs the aid of some editing service.

Author Response

- We are grateful to the reviewer for having carefully evaluated our article and for the useful advice aimed at improving it, both in form and in content.

- Because (after the first round) some doubts remained about linguistic correctness, the reviewer strongly suggested further correction / evaluation of an editing service. We appreciated the reviewer's comment for improving our manuscript. We requested the assistance of the MDPI language editing service. The text has therefore been completely revised and corrected by MDPI service, and is now considerably improved. All changes are highlighted in the draft, as per editorial guidelines.

- We are confident that these changes will satisfy the reviewer. A copy of the proofreading certificate from the MDPI language service is available from the corresponding author on reasonable request.

Round 3

Reviewer 2 Report

No comments for authors